# COINs: Model-based Accelerated Inference for Knowledge Graphs

## Abstract

We introduce COmmunity INformed graph embeddings (COINs), for accelerating link prediction and query answering models for knowledge graphs. COINs employ a community-detection-based graph data augmentation procedure, followed by a two-step prediction pipeline: node localization via community prediction and then localization within the predicted community. We describe theoretically justified criteria for gauging the applicability of our approach in our setting with a direct formulation of the reduction in time complexity. Additionally, we provide numerical evidence of superior scalability in model evaluation cost (average reduction factor of $6.413 \pm 3.3587$ on a single-CPU-GPU machine) with admissible effects on prediction performance (relative error to baseline $0.2657 \pm 0.3279$ on average).

## 1 Introduction

Knowledge graphs are a type of database where the general structure is a *network* of *entities*, their semantic *types*, *properties*, and *relationships*; relations are *flexible* through the use of *abstract classes* to represent real-world data from potentially multiple topical domains (Ehrlinger & Wöß, 2016; Hogan et al., 2020). Organizations often use knowledge graphs to organize and analyze relational data. However, scalability becomes a challenge as the graphs can become very large and require significant computational resources. This is particularly challenging for smaller organizations.

Previous research has already proposed several potential solutions. PyTorch-BigGraph (Lerer et al., 2019), and DistDGL (Zheng et al., 2020) are distributed-algorithm-system frameworks that can scale the implementation of graph neural networks with distributed execution over billion-scale graphs. Both approaches involve *splitting the enormous graph into disjoint subsets*, to employ distributed training by directing data from different subsets to different machines. SMORE (Ren et al., 2021) on the other hand, focuses on model parameter distribution over a cluster, while also employing a more efficient algorithm for sampling training data. tf-GNN (Ferludin et al., 2022) mainly focuses on accelerating the sampling of k-hop neighborhoods of nodes during training by distributing it over multiple machines.

However, all methods for achieving better scalability are inapplicable to all application scenarios. It is assumed that nevertheless, a minimal cluster size of machines is always available, and any graph partitioning rarely accounts for preserving graph properties. In a low resource availability scenario, even the model's evaluation is a hurdle with big data, which has not been considered in the literature. The evaluation procedure for top-performing knowledge graph representation models was either also sped up with data parallelization over a large cluster of machines (Cen et al. (2019)) or the evaluation metrics are only approximated with a subset of the graph (Ren et al. (2021)).

As such, in this work, we will present COINs (**CO**mmunity **IN**formed graph embedding**s**), our approach for achieving scalable knowledge graph model optimization, inference, and evaluation in the low-resource setting. Before explicitly stating our contributions below, we will first establish the background and introduce the preliminaries.

## 2 BACKGROUND

### 2.1 PROBLEM AND CHALLENGES

Performing knowledge graph inference requires solving link prediction and query answering.

**Definition 1.** *Given a dataset $\mathcal{D} = \{(h_i, r_i, t_i)\}_{i=1}^N$ of entity-relation-entity triplets and labels $\{y_i\}_{i=1}^N$ indicating whether each triplet was extracted from the knowledge graph or randomly formed, a **link predictor** $\ell : V \times R \times V \to \mathbb{R}$ is a model that aims to predict the label $y$ of each triplet $(h, r, t)$ by scoring it according to how likely it is to have originated from the knowledge graph.*

Link predictors commonly are composed of entity embedding models, in order to obtain high-quality feature vectors to pass onto a classifier.

**Definition 2.** *Given a dataset $\mathcal{D} = \{(h_i, r_i)\}_{i=1}^N$ of entity-relation queries and answer entities $\{t_i\}_{i=1}^N$ extracted from the knowledge graph, **single-hop query answering** entails predicting the answer entity $t$ given the query information $(h, r)$ as input. Classically, this is operationalized by forming the set $\hat{\mathcal{D}}_i = \{(h_i, r_i, \hat{t}_j)\}_{j=1}^{|V|}$ by combining the query with every possible answer, then the predicted answer is the one that maximizes the link predictor's score: $\hat{t}_i = \mathrm{argmax}_{\hat{t}_j \in V} \ell(h_i, r_i, \hat{t}_j)$.*

Binary classification evaluation metrics are utilized for the link prediction task. On the other hand, after obtaining the predicted *ranks* $\{\rho_i\}_{i=1}^N$ of the evaluation triplet scores (index of the score of the true triplet in an ordered list of scores), one can compute query answering aggregate metrics:

- **Hits@k** $= \frac{1}{N} \sum_{i=1}^N \mathbb{I}(\rho_i \leq k)$;
- **MRR** $= \frac{1}{N} \sum_{i=1}^N \frac{1}{\rho_i}$.

The query answering task requires evaluating the link prediction model multiple times for *each* of the $N$ data samples. This task is computationally intensive and the time complexity is determined by the number of node embeddings that need to be computed during evaluation. The largest contributor to the complexity is thus the size of the graph, $|V|$. Our work aims to significantly reduce this factor.

### 2.2 PRELIMINARIES

Mathematically we can represent knowledge graphs with attributed heterogeneous networks:

**Definition 3.** *An **Attributed HEterogeneous Network (AHEN)** (Liu et al., 2019) is a graph where each entity (node) in a set $V$ has numerical attributes and each relation (edge) in a set $E$ has an associated type label. Formally, first a finite set of **relation types** is defined $R = \{r_1, \ldots, r_Q\}$, and the relation labeling is represented through the mapping $t_R : E \to R$. Given this mapping we define the **set of relation triplets**: $E_R = \{(h, r, t) | h, t \in V, (h, t) \in E, r = t_R((h, t)) \in R\}$. Entity/node attributes can be collected e.g. in a matrix $X \in \mathbb{R}^{|V| \times T}$, with $T$ as the feature dimension. Usually, the nodes $h$ and $t$ in the triplets are referred to as the **head** and **tail** of the relations, respectively. The attributed heterogeneous network is the new collection $(V, E_R, R, t_R, X)$.*

Graphs are discrete structures, while the majority of machine learning models work exclusively with real-valued continuous inputs. Therefore, we have to construct a mapping from the graph elements to a numeric vector space, such that the output vectors encode the graph structure in the form of algebraic constraints.

**Definition 4.** *Given a knowledge graph $G = (V, E_R, R, t_R, X)$, in general one defines a **knowledge graph embedding model** as the collection of an entity embedding model $f_V : V \to \mathcal{E}$, relation embedding model $f_R : R \to \mathcal{R}$ and a scoring function $\mathcal{S} : \mathcal{E} \times \mathcal{R} \times \mathcal{E} \to \mathbb{R}$, where $D$ is the embedding dimension and $\mathcal{E}, \mathcal{R} \subseteq \mathbb{R}^D$ or $\mathbb{C}^D$. $f_V$ and $f_R$ are parametrized, with their parameters set to values such that $\forall (v, r, u) \in E_R, \mathcal{S}(f_V(v), f_R(r), f_V(u))$ is optimal.*

*The outputs $e_v = f_V(v)$ and $e_r = f_R(r)$ are referred to as entity and relation embeddings.*

Most graph embeddings are optimized using a contrastive modification of the scoring function:

**Definition 5.** *Given a single **positive** triplet $(h, r, t)$ and a set of **negative** triplets $\{(h'_i, r, t'_i)\}_{i=1}^{m}$ (relations not present in the knowledge graph), the **contrastive embedding loss** takes the common form (with $\sigma$ denoting the sigmoid function):*

$$\mathcal{L}(e_h, e_r, e_t) = -\log \sigma(\mathcal{S}(e_h, e_r, e_t)) - \frac{1}{m}\sum_{i=1}^{m}\log \sigma(-\mathcal{S}(e_{h'_i}, e_r, e_{t'_i})) \tag{1}$$

## 2.3 CONTRIBUTIONS

Previous work on knowledge graph inference acceleration (Lerer et al., 2019; Zheng et al., 2020; Ren et al., 2021) focuses on *model-free* solutions relying on hardware and software for distribution and parallelization. On the other hand:

I. COINs are a *model-based* approach that performs a novel integration of *community structure information* into optimization and inference, to achieve significant acceleration even with limited resources;

II. The acceleration potential is exactly derived, with theoretical and empirical arguments for which community structures are optimal.

### 2.3.1 SCALABLE QUERY ANSWERING EVALUATION

Assume that we possess a mapping $c : V \rightarrow \mathcal{C}$, assigning knowledge graph entities into one of $|\mathcal{C}| = K \leq |V|$ groups, similar to previous work (Lerer et al., 2019; Zheng et al., 2020). With $c$ we construct an alternative two-step query answering procedure:

1. Map the input $(h, r)$ to $(c(h), r)$ and predict $c(t)$ by scoring all $K$ possible answer groups;
2. Search for the correct $t$ only among the entities $\hat{t}$ with $c(\hat{t}) = c(t)$.

### 2.3.2 OPTIMAL PARTITION STRUCTURE

First note that the mapping $c$ induces a *disjoint partition* of the entity set $V$. Namely, if we construct the sets $C_k = \{v \in V | c(v) = k\}$, $k \in \mathcal{C}$, then $\bigcup_{k=1}^{K} C_k = V$ and $\forall i, j \in \mathcal{C}, i \neq j, C_i \cap C_j = \emptyset$.

We will now demonstrate that by analyzing in detail the computational complexity of our new evaluation method, one can easily assess the quality of any such disjoint partition.

**Proposition 1.** *Let $|C_k|$ denote the size of node group $k$ and $|E_k^{test}| = |\{(h, r, t) \in \mathcal{D}_{test}, c(t) = k\}|$ denote the number of evaluation relations with tail nodes in node group $k$. The number of embedding vectors computed in total during evaluation has the following form and bounds:*

$$2N\sqrt{|V|} \leq \sum_{k=1}^{K}(K + |C_k|)|E_k^{test}| \leq N(|V| + 1) \tag{2}$$

*Proof.* The time complexity of the first node group prediction step of the new evaluation procedure, is proportional to the number of groups $K$, as this is the number of embeddings required. On the other hand, the second step requires as many model evaluations as the number of nodes in the group, $|C_k|$. Each node group is represented by $|E_k^{\text{test}}|$ edges in the test data, and for each edge the total is thus $K + |C_k|$. By summing over all groups $k$ one obtains the provided exact complexity.

Regarding bounding the expression $\sum_{k=1}^{K}(K + |C_k|)|E_k^{\text{test}}|$, let's first assume that we have fixed the number of groups $K$ to some value in $\{1, \ldots, |V|\}$ and we're aiming to optimize the distribution of nodes and evaluation edges across groups for this fixed $K$. Using the KKT theorem, one can prove that extremal configurations only occur when all groups are of equal size and/or all groups are represented with an equal amount of edges in the evaluation set. Proposition 3 gives the details.

It is easier to strive towards $|C_k| \approx \frac{|V|}{K}$, however, in any case, the number of node embedding computations will then equal $N\left(K + \frac{|V|}{K}\right)$. To conclude, we prove that the value of $K$ is what decides whether the lower or upper bound stated above is achieved. Proposition 4 gives the details. $\qquad \square$

### 2.3.3 EDGE LOCALITY

The possibility of quadratically reducing the overall computation time is grounds for sufficient motivation to obtain a partitioning that splits the node set into $K = O(\sqrt{|V|})$ equally sized pieces, as its first important property. However, to promise to minimally affect the baseline performance, we must consider the partitioning quality from the perspective of preserving the relational structure. What previous works often fail to consider is that an arbitrary assignment might be *arbitrarily non-local*.

Formally, if we construct the set

$$V^* = \{v \in V | \exists (v, r, u) \in E_R \lor \exists (u, r, v) \in E_R, c(u) \neq c(v)\}$$

of entities that participate in between-group relations, then we must have $|V^*| \leq |V|$ be minimal. In this case, the search for the true answer to a query will be most often localized to a small neighborhood in the knowledge graph, a significantly easier problem. Equivalently, one prefers a mapping $c$ such that more relations are present *within* the entity sets $C_k$ than *between* the sets.

Unfortunately, achieving good localization is an NP-hard problem, but nevertheless, research in the fields of *community detection* and *minimal cuts* for graphs have yielded extremely efficient heuristic algorithms. Such is the Leiden algorithm for community detection by Traag et al. (2019), the current state-of-the-art, based on heuristic maximization of the Constant Potts Model (CPM) (Traag et al., 2011) version of the modularity score (Blondel et al., 2008), with further manual refinement to guarantee well-connected communities. The single "resolution" hyperparameter of the algorithm is the only degree of freedom controlling the community assignment quality. Our work represents a novel application of Leiden communities to knowledge graph inference. On the other hand, the METIS algorithm for graph partitioning (Karypis & Kumar, 1998), employed also by Zheng et al. (2020), strives towards minimal graph cuts. For some experiments, we considered it as an alternative to Leiden in order to estimate the effect of the choice of graph-splitting method.

## 3 METHODOLOGY

### 3.1 COMMUNITY DETECTION

Given a knowledge graph $(V, E_R, R, t_R, X)$, the first step we employ is to execute the Leiden community detection algorithm to obtain a good node-to-community mapping $c : V \to \mathcal{C}$. We observed that, in practice, if the graph is sufficiently sparse, i.e. $|E_R| << |R| \cdot |V|^2$, with the resolution parameter of the algorithm simply set to $\frac{1}{|V|}$ one achieves decent performance without the need for further validation. Obtaining quality community assignments for dense graphs, however, is very challenging, and validation of the resolution hyperparameter is recommended. Additionally, one can merge very small weakly connected components into larger communities to achieve a better balance of community sizes.

For our COINs technique, based on the obtained community assignment map $c$ we extract further discrete information of great utility for efficient training and evaluation. Using $c$, one can coarsen $(V, E_R)$ into a *community-level interaction graph*:

$$V^C = \mathcal{C}, E_R^C = \{(c(u), r, c(v)) | (u, r, v) \in E_R\}$$

In addition, one can compute the *inter-community mapping* $z^* : V \to V_\omega^*, V_\omega^* = V^* \cup \{\omega\}$:

$$z^*(v) = \begin{cases} v, & \text{if } v \in V^* \\ \omega, & \text{otherwise} \end{cases}$$

Figure 1 provides a visualization of our community-based graph preprocessing on a small example.

### 3.2 COINS TRAINING & EVALUATION

Let $\mathcal{F}$ denote the class of knowledge graph embedding models whose scalability we wish to improve, and $\mathcal{L} : \mathcal{E} \times \mathcal{R} \times \mathcal{E} \times \{0, 1\} \to \mathbb{R}$ be the loss function required to train such models. COINs require the optimization of the following parameterized embedders sampled from $\mathcal{F}$:

- **Community embedder** $f_C : \mathcal{C} \times R \times \mathcal{C} \to \mathcal{E} \times \mathcal{R} \times \mathcal{E}$;

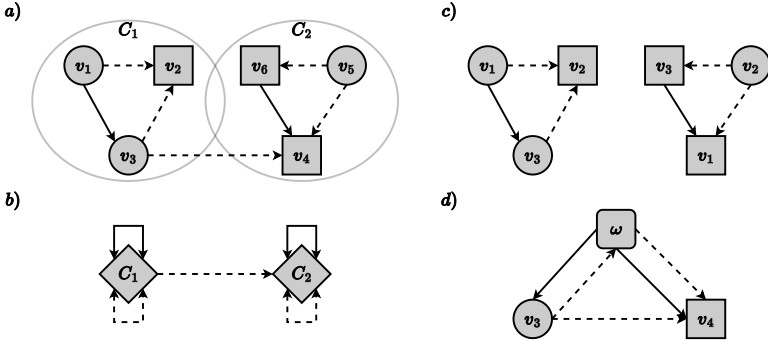

Figure 1: a) Example knowledge graph with $|V| = 6, |T| = 2, |R| = 2, K = 2$. b) Obtained community-level graph. c) Intra-community re-indexed nodes and relevant edges. d) Inter-community mapped nodes and relevant edges.

- **Intra-community embedders** $f_k : C_k \times R \times C_k \to \mathcal{E} \times \mathcal{R} \times \mathcal{E}, \forall k \in \mathcal{C}$;
- **Inter-community embedder** $f_* : V_\omega^* \times R \times V_\omega^* \to \mathcal{E} \times \mathcal{R} \times \mathcal{E}$.

Additional parameters are: a **node feature embedding matrix** $W \in \mathbb{R}^{D \times T}$ and **final node/relation embedding weights** $w \in \mathbb{R}^3, w_R \in \mathbb{R}^2$. Algorithm 1 details the knowledge graph triplet representation steps for COINs, utilizing these constructed sub-models.

---

**Algorithm 1** COINs knowledge graph representation

---
1: **input** triplet $(h, r, t) \in E_R$, triplet label $y \in \{0, 1\}$, community assignment $c : V \to \mathcal{C}$, inter-community map $z^* : V \to V_\omega^*$, loss weight hyperparameter $\alpha \in (0, 1)$
2: $e_h^{(c)}, e_r^{(c)}, e_t^{(c)} = f_C(c(h), r, c(t))$
3: **if** $c(h) = c(t)$ **then**
4:     $e_h^*, e_r^*, e_t^* = f_{c(h)}(h, r, t)$
5: **else**
6:     $e_h^*, e_r^*, e_t^* = f_*(z^*(h), r, z^*(t))$
7: **end if**
8: $e_h = \langle \text{Softmax}(w), [Wx_h, e_h^{(c)}, e_h^*] \rangle$
9: $e_r = \langle \text{Softmax}(w_R), [e_r^{(c)}, e_r^*] \rangle$
10: $e_t = \langle \text{Softmax}(w), [Wx_t, e_t^{(c)}, e_t^*] \rangle$
11: **return** $(1 - \alpha)\mathcal{L}(e_h^{(c)}, e_r^{(c)}, e_t^{(c)}; y) + \alpha\mathcal{L}(e_h, e_r, e_t; y)$

---

This construction has several desirable properties:

1. Between-community relations are learned separately, allowing greater generalization;

2. During the forward and backward pass always embedding matrix parts of size much less than $O(|V|D)$ are updated, yielding much lower per iteration time and memory complexity;

3. The community representation learning will be performed jointly with the node-level, due to the final embedding refinement linking them both, allowing for greater adaptivity;

4. The convexity of the final embedding refinement will preserve training stability.

Observe that the total number of COINs node embeddings required in memory in total for a graph is $K + \sum_{k=1}^{K} |C_k| + |V^*| + 1 = K + |V| + |V^*| + 1$, compared to just $|V|$ node embeddings in the baseline. Thus, using a partitioning that keeps $|V^*| << |V|$ has scalability-related benefits as well.

Let now $\ell_C : \mathcal{C} \times R \times \mathcal{C} \to \mathbb{R}$ denote a learned scoring model for relations between communities, obtained after extending $f_C$ into a classifier, while $\ell_V : V \times R \times V \to \mathbb{R}$ denoting the classical scoring model for node relations, now a classifier utilizing the final node embeddings $(e_h, e_r, e_t)$

from COINs as features. Then, one can implement the proposed novel scalable query answering procedure via Algorithm 2.

---

**Algorithm 2** COINs query answering evaluation

---

1: **input** test set $\mathcal{D} = \{(h_i, r_i, t_i)\}_{i=1}^N$, link scorers $\ell_C, \ell_V$
2: **for** $i \leftarrow 1$ to $N$ **do**
3:      $s_C = \{\ell_C(c(h_i), r_i, k)\}_{k=1}^K$
4:      $\rho_C^{(i)} = \text{Rank}(c(t_i), s_C)$
5:      $F_i = \sum_{1 \le k < \rho_C^{(i)}} |\{v | v \in V, \text{Rank}(c(v), s_C) = k\}|$
6:      $s_V = \{\ell_V(h_i, r_i, v_j)\}_{v_j \in C_{c(t_i)}}$
7:      $\rho_{c(t_i)}^{(i)} = \text{Rank}(t_i, s_V)$
8:      $\rho^{(i)} = F_i + \rho_{c(t_i)}^{(i)}$
9: **end for**
10: **return** $\{\rho^{(i)}\}_{i=1}^N$

---

### 3.3 TRADEOFF ANALYSIS

The primary limitation of the COINs evaluation procedure is that incorrect community prediction can result in many nodes being returned as false positives. This conditional nature of the prediction is the main reason for which to expect the original embedder's performance to be the maximum achievable. However, as community ranks approach 1, for large graphs we can much more quickly obtain model predictions with lesser and lesser drawbacks. Thus, the following Proposition 2 quantifies under which conditions the application of COINs can be justified, by theoretically analyzing the tradeoff between the possible scalability benefits and performance losses.

**Proposition 2.** *Let $H_k \in \{0, 1, \dots\}$ be the r.v. counting how many edges need to be evaluated before a correct hit in the top $k$ results is achieved, and let $\varepsilon \in \mathbb{R}$ denote the relative error in Hits@k to the baseline incurred after training and evaluating the model with COINs. Then, when assuming that $H_k \sim Geom(Hits@k(1 - \varepsilon))$, the application of COINs is justified if:*

$$\varepsilon < 1 - \frac{\sum_{k=1}^K (K + |C_k|)|E_k^{test}|}{N|V|} \Leftrightarrow \frac{N|V|}{\sum_{k=1}^K (K + |C_k|)|E_k^{test}|} > \frac{1}{1 - \varepsilon} \tag{3}$$

*Proof.* Let $T = |V|$ denote the baseline evaluation complexity per edge, while $T'$ the COINs computation cost per edge, given in Proposition 1 (up to dividing by $N$). The better model is deemed to be the one with lower overall expected cost up until a correct hit:

$$\mathbb{E}[T'H_k'] < \mathbb{E}[TH_k] \Leftrightarrow \frac{T'}{Hits@k(1 - \varepsilon)} < \frac{|V|}{Hits@k} \Leftrightarrow \varepsilon < 1 - \frac{NT'}{N|V|}$$

$\square$

### 3.4 EXPERIMENT SETUP

#### 3.4.1 INTEGRATION WITH KNOWLEDGE GRAPH REPRESENTATION MODELS

Due to the universality of our COINs technique for scaling up knowledge graph model evaluation, we can integrate it with many established knowledge graph embedding models, described in Table 1 below. Each performs contrastive learning, i.e., minimization of equation 1.

#### 3.4.2 DATA

Our approach was tested on three knowledge graph datasets used classically for knowledge graph representation research:

- **FB15k-237** (Toutanova & Chen, 2015): sample of the Freebase knowledge base;
- **WN18RR** (Dettmers et al., 2018): a subset of the WordNet ontology;
- **NELL-995** (Xiong et al., 2017): 995th iteration of the NELL knowledge reasoning system.

Table 1: Knowledge graph embedders summary. Left to right: node embedding space, relation embedding space, embedding parameter constraints, score function.

| Model | $\mathcal{E}$ | $\mathcal{R}$ | Constraints | $\mathcal{S}(e_h, e_r, e_t)$ |
|---|---|---|---|---|
| **TransE** (Bordes et al., 2013) | $\mathbb{R}^D$ | $\mathbb{R}^D$ | $\|\|e_h\|\| = \|\|e_t\|\| = 1$ | $\gamma - \|\|e_h + e_r - e_t\|\|$ |
| **DistMult** (Yang et al., 2014) | $\mathbb{R}^D$ | $\mathbb{R}^D$ | $\|\|e_h\|\| = \|\|e_t\|\| = 1$ | $\langle e_h \odot e_r, e_t \rangle$ |
| **ComplEx** (Trouillon et al., 2016) | $\mathbb{C}^D$ | $\mathbb{C}^D$ | None | $\mathrm{Re}(\langle e_h \cdot e_r, \bar{e}_t \rangle)$ |
| **RotatE** (Sun et al., 2019) | $\mathbb{C}^D$ | $\mathbb{C}^D$ | $\forall d, \left\| e_r^{(d)} \right\| = 1$ | $\gamma - \|\|e_h \cdot e_r - e_t\|\|$ |

### 3.4.3 TRAINING SETUP

We ran the COINs training and evaluation procedures on all combinations of knowledge graph datasets and embedding algorithms, each started from one of 5 random seeds. Positive and negative sample triplets are obtained online during training to form mini-batches, while fixed validation and testing datasets are built by sampling 5 negative samples for every given evaluation edge. In both cases, the efficient bi-directional rejection sampling algorithm of SMORE (Ren et al., 2021) is applied. The parameter optimization is performed via the Adam optimization algorithm (Kingma & Ba, 2015), with $\ell_2$ parameter regularization. Training loss absolute difference threshold plus patience counter incremented on validation loss non-decrease are employed as early stopping criteria.

## 4 RESULTS & DISCUSSION

### 4.1 COMMUNITIES & SCALABILITY

Table 2 summarizes the three knowledge graphs via their basic statistics, as well as those of the community partitioning that we managed to obtain. We present the time complexity improvements via the *acceleration* factor: ratio of node embeddings computed during evaluation between COINs and the baseline (ratio influencing the Proposition 2 bound as well). On the other hand, the effects on memory complexity can be analyzed via the ratio of total composing node embeddings between COINs and baseline models, referenced as the *overparametrization* factor.

Table 2: Knowledge graph datasets summary. Left to right: number of nodes, edges, node features, edge types, communities; acceleration factor (decrease in evaluation time complexity, higher is better), overparametrization factor (increase in memory complexity, lower is better).

| Dataset | $\|V\|$ | $\|E_R\|$ | $T$ | $\|R\|$ | $K$ | Acceleration | Overparametrization |
|---|---|---|---|---|---|---|---|
| FB15k-237 | 14541 | 310116 | 1 | 237 | 1025 | x 4.3664 | x 1.9926 |
| WN18RR | 41105 | 93003 | 5 | 11 | 88 | x 4.5833 | x 1.156 |
| NELL-995 | 75492 | 154213 | 269 | 200 | 275 | x 10.2892 | x 1.0687 |

We note the inability to bring down $K$ and $\|V^*\|$ for the FB15k-237 graph due to its high edge density, yielding the worst scalability effects. Nevertheless, we provide evidence that through elementary validation of the resolution hyperparameter one can optimize the scalability according to desired preferences, due to a surprisingly strong correlation of the factors with the resolution value. Figure 2 illustrates this through a joint plot of both factors, and one can notice the common pattern across the datasets: there's a critical point for optimal acceleration, while overparametrization simply increases with the resolution.

### 4.2 PERFORMANCE & FEASIBILITY

Table 3 contains our query answering results and the comparison with the baselines, obtained from running the implementation by Sun et al. (2019)[1]. One can observe that on average, our novel method for knowledge graph embedding training and evaluation does not reduce greatly the prediction performance promised by the baselines. In fact, in a few cases, we seem to improve upon the results and the RotatE algorithm seems to be the most compatible with COINs.

---

[1]Available at `https://github.com/DeepGraphLearning/KnowledgeGraphEmbedding`

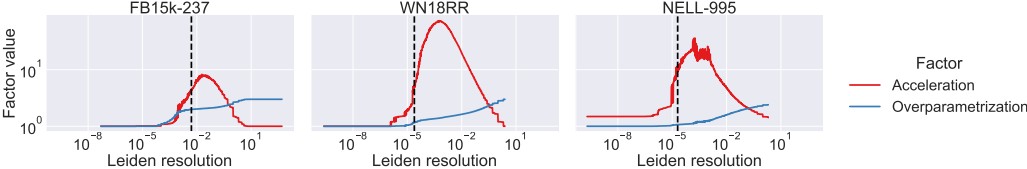

Figure 2: Dependence of time and memory scalability (acceleration and overparametrization factors) on the value of the resolution hyperparameter of the Leiden community detection algorithm. Left to right: different datasets. Chosen hyperparameter values that yielded optimal balance between scalability and performance for each dataset annotated via vertical lines.

Table 3: All computed query answering metrics (higher is better): comparison of our results with COINs training and evaluation to baselines with equal hyperparameters. Highlighted values indicate the superiority of COINs or a relative error lower than 10%.

| Dataset | Model | Value | C-H@1 | H@1 | H@3 | H@10 | MRR |
|---|---|---|---|---|---|---|---|
| | TransE | Baseline | - | 0.142 | 0.240 | 0.369 | 0.218 |
| | | COINs | 0.676 ± 0.000 | 0.078 ± 0.003 | 0.136 ± 0.005 | 0.245 ± 0.007 | 0.132 ± 0.004 |
| | DistMult | Baseline | - | 0.254 | 0.377 | 0.527 | 0.344 |
| | | COINs | 0.291 ± 0.045 | 0.038 ± 0.007 | 0.074 ± 0.012 | 0.132 ± 0.012 | 0.068 ± 0.009 |
| FB15k-237 | ComplEx | Baseline | - | 0.278 | 0.404 | 0.552 | 0.369 |
| | | COINs | 0.975 ± 0.007 | **0.333** ± **0.007** | **0.477** ± **0.007** | **0.626** ± **0.007** | **0.431** ± **0.006** |
| x4.4 speed-up | RotatE | Baseline | - | 0.282 | 0.430 | 0.584 | 0.383 |
| | | COINs | 0.944 ± 0.001 | **0.295** ± **0.003** | **0.416** ± **0.004** | **0.552** ± **0.004** | **0.381** ± **0.003** |
| | TransE | Baseline | - | 0.019 | 0.241 | 0.416 | 0.160 |
| | | COINs | 0.941 ± 0.007 | **0.199** ± **0.006** | **0.311** ± **0.008** | **0.436** ± **0.012** | **0.278** ± **0.008** |
| | DistMult | Baseline | - | 0.399 | 0.452 | 0.489 | 0.433 |
| | | COINs | 0.997 ± 0.001 | 0.176 ± 0.063 | 0.305 ± 0.086 | **0.423** ± **0.077** | 0.261 ± 0.071 |
| WN18RR | ComplEx | Baseline | - | 0.426 | 0.479 | 0.526 | 0.462 |
| | | COINs | 0.999 ± 0.000 | 0.297 ± 0.020 | 0.394 ± 0.017 | 0.466 ± 0.010 | 0.358 ± 0.018 |
| x4.6 speed-up | RotatE | Baseline | - | 0.442 | 0.491 | 0.538 | 0.476 |
| | | COINs | 0.998 ± 0.000 | **0.436** ± **0.001** | **0.510** ± **0.003** | **0.586** ± **0.004** | **0.487** ± **0.001** |
| | TransE | Baseline | - | 0.230 | 0.368 | 0.448 | 0.312 |
| | | COINs | 0.971 ± 0.000 | 0.150 ± 0.011 | 0.244 ± 0.012 | 0.356 ± 0.019 | 0.218 ± 0.009 |
| | DistMult | Baseline | - | 0.315 | 0.434 | 0.555 | 0.395 |
| | | COINs | 0.906 ± 0.033 | 0.062 ± 0.018 | 0.129 ± 0.025 | 0.333 ± 0.013 | 0.127 ± 0.022 |
| NELL-995 | ComplEx | Baseline | - | 0.362 | 0.538 | 0.635 | 0.466 |
| | | COINs | 0.996 ± 0.001 | 0.097 ± 0.037 | 0.193 ± 0.070 | 0.296 ± 0.066 | 0.161 ± 0.045 |
| | RotatE | Baseline | - | 0.433 | 0.520 | 0.562 | 0.482 |
| x10.3 speed-up | | COINs | 0.996 ± 0.000 | 0.304 ± 0.015 | **0.491** ± **0.023** | **0.604** ± **0.016** | **0.412** ± **0.018** |

The community-graph Hits@1 metric provides further insight into our query answering performance, obtained after analysing separately the metrics for each of the two prediction steps. Query answering over the community interaction graphs yielded near-perfect responses, except for the case of the FB15k-237 knowledge graph, whose community-interaction graph proved difficult to model for the simpler TransE and DistMult methods. In these cases where the performance on community prediction is not close to ideal, the overall performance will also be directly negatively impacted as expected, due to the conditional nature of the prediction procedure.

However, a relevant comparison of the exact numbers showcased in Table 3 is difficult to perform without the aid of an additional scalability-dependent investigation of the applicability of our proposed approach. Thus, Figure 3 proposes a manner for visually verifying the satisfaction of the relative error condition from Proposition 2. In the upper row, we observe the relative error in overall Hits@k metrics against the acceleration factor. We observe that most points lie in the feasibility region, except for the weak performance of TransE and DistMult on the FB15k-237 graph, which we commented on previously. However, as shown in the bottom row, where we remove the impact of the community prediction performance and only focus on node Hits@k, we observe that the inequality is now satisfied even for these hardest scenarios.

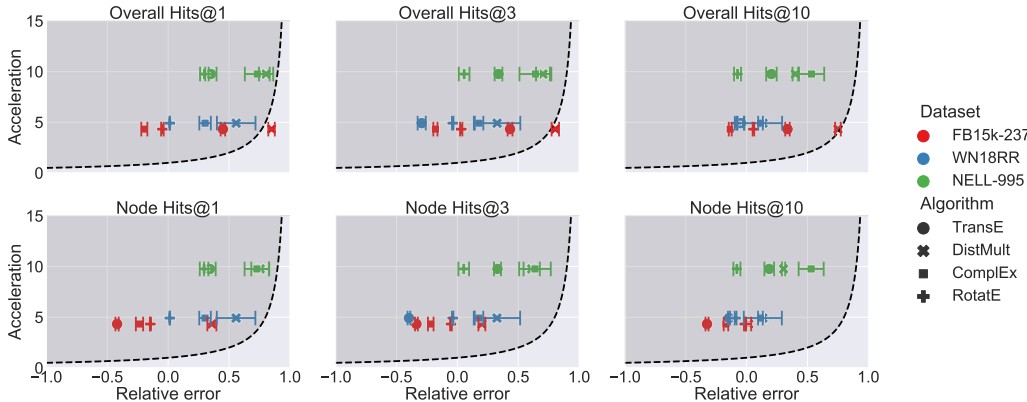

Figure 3: Plot of relative error in Hits@k metrics against acceleration factor. The feasible region (Proposition 2) is the one shaded. Top vs. bottom: COINs metric values after both steps vs. just the second step. We observe one case where $\approx$10.3 speed-up is paired with $\approx$75% loss in performance, while also $\approx$10% improvement in another. The choice of embedding algorithm thus remains crucial.

## 5 CONCLUSION

We introduced COINs, a method for accelerating link prediction and query answering models for knowledge graphs, to be applicable even in low computational resource settings. COINs featured a two-step prediction procedure: node localization via community prediction and within-community node prediction.

This first node localization step was found to be very impactful on overall model performance. We theoretically and empirically elaborated that the quality of the community structure of the knowledge graph has a broad-reaching influence on the possible scalability improvements provided by our method, as well as its prediction performance compared to baselines. As such, one can conclude that before selecting COINs to accelerate the embedding of a knowledge graph, it is important to study and evaluate its community structure.

As relevant future work, we propose a continuation of the investigation into the applicability of COINs when integrated into knowledge graph representation models of greater parameter complexity and prediction power, employing graph convolution or graph attention (Cen et al., 2019; Shang et al., 2019; Nathani et al., 2019). In addition, one can focus on attempting the integration of COINs with algorithms for multi-hop knowledge graph reasoning (Ren et al., 2020; Ren & Leskovec, 2020), as experiments so far focused only on the single-hop query answering problem.

**Reproducibility Statement**   Appendix D provides helpful details on how to reproduce our results, by listing the hardware and software utilized, as well as providing a link to a repository with the full code. Therein, one can find the instructions on how to rerun our experiments and regenerate the paper's tables and figures.

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

## A  ADDITIONAL PROOFS

**Proposition 3.** *Let* $\{a_k\}_{k=1}^K = \{|C_k|\}_{k=1}^K$ *and* $\{b_k\}_{k=1}^K = \{|E_k^{test}|\}_{k=1}^K$ *denote the learnable parameters. Then with the constraints* $\sum_{k=1}^K a_k = |V|, \sum_{k=1}^K b_k = N, \forall k, a_k > 0, b_k \geq 0$, *the only extremal points of* $g(\{a_k\}, \{b_k\}) = \sum_{k=1}^K (K + a_k)b_k$ *are when* $\forall k, a_k^* = \frac{|V|}{K}$ *and/or* $\forall k, b_k^* = \frac{N}{K}$, *with the extreme value of* $g(\{a_k^*\}, \{b_k^*\}) = N\left(K + \frac{|V|}{K}\right)$.

*Proof.* The Lagrangian of the optimization problem is the following:

$$L(\{a_k\}, \{b_k\}) = \sum_{k=1}^K (K + a_k)b_k + \lambda_1\left(\sum_{k=1}^K a_k - |V|\right) + \lambda_2\left(\sum_{k=1}^K b_k - N\right) - \sum_{k=1}^K \mu_k a_k - \sum_{k=1}^K \nu_k b_k$$

From the stationarity KKT conditions:

$$\forall k, \frac{\partial L}{\partial a_k} = 0 \Leftrightarrow \forall k, b_k + \lambda_1 - \mu_k = 0 \Leftrightarrow \forall k, b_k = \mu_k - \lambda_1$$

$$\forall k, \frac{\partial L}{\partial b_k} = 0 \Leftrightarrow \forall k, a_k + K + \lambda_2 - \nu_k = 0, \Leftrightarrow \forall k, a_k = \nu_k - \lambda_2 - K$$

Now, by the primal equality constraints:

$$\sum_{k=1}^{K} a_k = |V| \Leftrightarrow \sum_{k=1}^{K} (\nu_k - \lambda_2 - K) = |V| \Leftrightarrow \lambda_2 = \frac{1}{K} \sum_{k=1}^{K} \nu_k - \frac{|V|}{K} - K$$

$$\Rightarrow \forall k, a_k = \nu_k - \frac{1}{K} \sum_{j=1}^{K} \nu_j + \frac{|V|}{K}$$

$$\sum_{k=1}^{K} b_k = N \Leftrightarrow \sum_{k=1}^{K} (\mu_k - \lambda_1) = N \Leftrightarrow \lambda_1 = \frac{1}{K} \sum_{k=1}^{K} \mu_k - \frac{N}{K}$$

$$\Rightarrow \forall k, b_k = \mu_k - \frac{1}{K} \sum_{j=1}^{K} \mu_j + \frac{N}{K}$$

Finally, by the complementary slackness KKT conditions, primal inequality constraints, and dual feasibility KKT conditions:

$$\sum_{k=1}^{K} \mu_k a_k = 0 \Leftrightarrow \forall k, \mu_k = 0 \Rightarrow \forall k, b_k^* = \frac{N}{K} \qquad \text{as } \forall k, a_k > 0, \mu_k \geq 0$$

$$\sum_{k=1}^{K} \nu_k b_k = 0 \Leftarrow \forall k, \nu_k = 0 \Rightarrow \forall k, a_k^* = \frac{|V|}{K} \qquad \text{as } \forall k, b_k \geq 0, \nu_k \geq 0$$

$\square$

**Proposition 4.** *With $K = \sqrt{|V|}$, one achieves the minimal evaluation cost of $2N\sqrt{|V|}$, while with $K \in \{1, |V|\}$ one achieves the maximum of $N(|V| + 1)$.*

*Proof.* Let $f(K) = N(K + \frac{|V|}{K})$. Then note that $f'(K) = N - \frac{N|V|}{K^2}$. From this, we find the only critical point of $f$ in $(1, |V|)$: $N - \frac{N|V|}{K^2} = 0 \Leftrightarrow K^2 = \frac{N|V|}{N} \Leftrightarrow K = \sqrt{|V|}$.

Now $f''(K) = \frac{2N|V|}{K^3}$. As $\forall K \in [1, |V|], f''(K) > 0$, $f$ is a convex function for all values of $K$. Thus, $\sqrt{|V|}$ will be both a local and global minimum, while the endpoints of the interval will be global maxima, as $\forall K \neq \sqrt{|V|}, f'(K) > 0$, i.e., $f$ increases away from the minimum in both directions. $\square$

## B  HYPERPARAMETERS

Table 4 lists the chosen values of the main hyperparameters influencing community detection, model architecture and optimization.

## C  ADDITIONAL RESULTS

### C.1  COMMUNITIES & SCALABILITY

Figure 4 illustrates an alternative method to optimize the scalability factors, through optimization of the cut size (number of inter-community edges) heuristic utilized by the METIS algorithm. For Leiden, one observes smooth curves with similar properties as in Figure 2, where we had the resolution parameter as the independent variable. For optimal acceleration, there seems to be a critical cut size,

Table 4: Hyperparameter configurations. Left to right: Leiden resolution, embedding dimension, contrastive loss margin, COINs loss weight, mini-batch size, number of negative samples per positive, total number of training samples, learning rate, and regularization weight.

| Dataset | Model | resolution | $D$ | $\gamma$ | $\alpha$ | $B$ | $m$ | $n$ | l.r. | $\lambda$ |
|---|---|---|---|---|---|---|---|---|---|---|
| FB15k-237 | TransE | $5 \cdot 10^{-3}$ | 100 | 1.0 | 0.5 | 256 | 128 | $2 \cdot 10^8$ | $10^{-3}$ | $10^{-6}$ |
| | DistMult | $5 \cdot 10^{-3}$ | 100 | n/a | 0.5 | 256 | 128 | $2 \cdot 10^9$ | $10^{-3}$ | $10^{-6}$ |
| | ComplEx | $5 \cdot 10^{-3}$ | 100 | n/a | 0.5 | 256 | 128 | $2 \cdot 10^9$ | $10^{-3}$ | $10^{-6}$ |
| | RotatE | $5 \cdot 10^{-3}$ | 100 | 9.0 | 0.5 | 256 | 128 | $2 \cdot 10^8$ | $10^{-3}$ | $10^{-6}$ |
| WN18RR | TransE | $2.4 \cdot 10^{-5}$ | 100 | 1.0 | 0.5 | 256 | 128 | $2 \cdot 10^8$ | $10^{-3}$ | $10^{-6}$ |
| | DistMult | $2.4 \cdot 10^{-5}$ | 100 | n/a | 0.5 | 256 | 128 | $2 \cdot 10^9$ | $10^{-3}$ | $10^{-6}$ |
| | ComplEx | $2.4 \cdot 10^{-5}$ | 100 | n/a | 0.5 | 256 | 128 | $2 \cdot 10^9$ | $10^{-3}$ | $10^{-6}$ |
| | RotatE | $2.4 \cdot 10^{-5}$ | 100 | 6.0 | 0.5 | 256 | 128 | $2 \cdot 10^8$ | $10^{-3}$ | $10^{-6}$ |
| NELL-995 | TransE | $2 \cdot 10^{-5}$ | 100 | 1.0 | 0.5 | 256 | 128 | $2 \cdot 10^8$ | $10^{-3}$ | $10^{-6}$ |
| | DistMult | $2 \cdot 10^{-5}$ | 100 | n/a | 0.5 | 256 | 128 | $2 \cdot 10^9$ | $10^{-3}$ | $10^{-6}$ |
| | ComplEx | $2 \cdot 10^{-5}$ | 100 | n/a | 0.5 | 256 | 128 | $2 \cdot 10^9$ | $10^{-3}$ | $10^{-6}$ |
| | RotatE | $2 \cdot 10^{-5}$ | 100 | 6.0 | 0.5 | 256 | 128 | $2 \cdot 10^8$ | $10^{-3}$ | $10^{-6}$ |

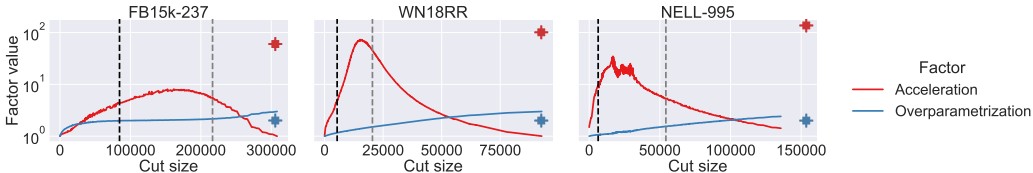

Figure 4: Dependence of time and memory scalability (acceleration and overparametrization factors) on the cut size (number of inter-community edges) of community partitions obtained by varying the resolution hyperparameter of the Leiden community detection algorithm. Left to right: different datasets. Cut values that yielded optimal balance between scalability and performance for each dataset annotated via black vertical lines. Gray vertical lines denote the cut size values obtained by the METIS algorithm, while the boxes with error bars denote the results from a batch of 100 random uniform community assignments.

while having more inter-community edges implied more parameters for $f_*$. Curiously, we observed the METIS algorithm to yield a non-minimal cut size and there are a lot of Leiden resolution values that achieve lower cut sizes. METIS clearly favors too large of an increase in parameter number. The distribution of the metrics over a batch of 100 random uniform community assignments yielded extremal values: the best acceleration, however the largest cut size and most parameters.

Figure 5 illustrates yet another alternative method to optimize the scalability factors, through optimization of the modularity heuristic utilized by the Leiden algorithm. In general, for Leiden, one observes both more acceleration and less overparametrization as modularity increases, however, as the plots display, we found this method to be more unstable than simply traversing the resolution hyperparameter space as before in Figure 2. We also note that the METIS algorithm produced slightly higher modularity only for the WN18RR dataset. Random community assignments again, naturally, yield extremal values.

## C.2 PERFORMANCE & FEASIBILITY

Table 5 provides further insight into our query answering performance, after analysing separately the metrics for each of the two prediction steps. We can observe how well the community Hits@1 score captures the influence of the accuracy in the community prediction on the overall performance. Namely, with some quick calculations, one can interestingly note that the overall COINs metric values can be closely approximated as the product of the community Hits@1 and respective scores for the within-community node prediction.

Figure 5: Dependence of time and memory scalability (acceleration and overparametrization factors) on the value of the modularity heuristic of the Leiden community detection algorithm. Left to right: different datasets. Modularity values that yielded optimal balance between scalability and performance for each dataset annotated via black vertical lines. Gray vertical lines denote the modularity values obtained by the METIS algorithm, while the boxes with error bars denote the results from a batch of 100 random uniform community assignments.

Table 6 contains all of our results on the link prediction task. We do not possess baselines for these results due to the limited scope of the related work, which focuses only on an evaluation of the query answering task. Thus, we cannot perform a similar comparison as before. Regardless, the relative ordering of the metric values across settings is consistent with the query answering performance discussed previously.

Figure 6 is our most detailed experiment, with the goal of merging together the validation of community assignments with the analysis of model feasibility, to investigate the tradeoff between achieved community quality, relative error in performance and acceleration/overparametrization. To facilitate this, for each dataset we end-to-end trained and evaluated the COINs-RotatE combination (as our most successful) with different sources of community assignments. Namely, we searched the region around the optimal value for the Leiden resolution hyperparameter to obtain worse communities, replaced Leiden with METIS for another case, and for a final run picked one random uniform community assignment.

As in Figures 2 and 4, we observe that the Leiden trajectories for all datasets have a critical point with an optimal trade-off between relative error and acceleration. We note that overall relative error, although unstable w.r.t. Leiden resolution, does not vary greatly and the main differences seem to lie in the acceleration values (although for FB15k-237 we manage to enter and exit the feasibility region by changing resolution). On the other hand, when considering only the performance in the second prediction step, relative error improves with Leiden resolution much faster. Note that, however, this is to be expected, as greater Leiden resolution implies smaller communities and thus, fewer possible answers to queries. We confirm again that overparametrization simply increases with greater resolution values and dominates the trajectory direction in the overall performance case. Thus, we confirm again that although the METIS and random communities might yield higher speed-ups, there are a lot of Leiden community assignments resulting in similar performance, and equally significant speed-ups but with significantly fewer parameters.

Table 5: Decomposition of COINs metrics into the overall values shown before and metrics for the second prediction step separately. Highlighted values indicate the superiority of COINs compared to the baseline or a relative error lower than 10%.

| Dataset | Model | Value | C-H@1 | H@1 | H@3 | H@10 | MRR |
|---|---|---|---|---|---|---|---|
| FB15k-237 | TransE | Overall | 0.676 ± 0.000 | 0.078 ± 0.003 | 0.136 ± 0.005 | 0.245 ± 0.007 | 0.132 ± 0.004 |
| | | Node | | **0.202 ± 0.002** | **0.321 ± 0.003** | **0.490 ± 0.005** | **0.296 ± 0.002** |
| | DistMult | Overall | 0.291 ± 0.045 | 0.038 ± 0.007 | 0.074 ± 0.012 | 0.132 ± 0.012 | 0.068 ± 0.009 |
| | | Node | | 0.162 ± 0.009 | 0.301 ± 0.011 | **0.518 ± 0.011** | 0.272 ± 0.009 |
| | ComplEx | Overall | 0.975 ± 0.007 | **0.333 ± 0.007** | **0.477 ± 0.007** | **0.626 ± 0.007** | **0.431 ± 0.006** |
| | | Node | | **0.344 ± 0.008** | **0.493 ± 0.009** | **0.645 ± 0.010** | **0.445 ± 0.008** |
| x4.4 speed-up | RotatE | Overall | 0.944 ± 0.001 | **0.295 ± 0.003** | **0.416 ± 0.004** | **0.552 ± 0.004** | **0.381 ± 0.003** |
| | | Node | | **0.323 ± 0.001** | **0.453 ± 0.003** | **0.593 ± 0.004** | **0.414 ± 0.002** |
| WN18RR | TransE | Overall | 0.941 ± 0.007 | **0.199 ± 0.006** | **0.311 ± 0.008** | **0.436 ± 0.012** | **0.278 ± 0.008** |
| | | Node | | **0.212 ± 0.004** | **0.336 ± 0.003** | **0.477 ± 0.003** | **0.299 ± 0.002** |
| | DistMult | Overall | 0.997 ± 0.001 | 0.176 ± 0.063 | 0.305 ± 0.086 | **0.423 ± 0.077** | 0.261 ± 0.071 |
| | | Node | | 0.176 ± 0.063 | 0.305 ± 0.086 | **0.424 ± 0.076** | 0.261 ± 0.071 |
| | ComplEx | Overall | 0.999 ± 0.000 | 0.297 ± 0.020 | 0.394 ± 0.017 | 0.466 ± 0.010 | 0.358 ± 0.018 |
| | | Node | | 0.297 ± 0.020 | 0.394 ± 0.017 | 0.467 ± 0.010 | 0.358 ± 0.018 |
| x4.6 speed-up | RotatE | Overall | 0.998 ± 0.000 | **0.436 ± 0.001** | **0.510 ± 0.003** | **0.586 ± 0.004** | **0.487 ± 0.001** |
| | | Node | | **0.436 ± 0.001** | **0.510 ± 0.003** | **0.586 ± 0.004** | **0.487 ± 0.001** |
| NELL-995 | TransE | Overall | 0.971 ± 0.000 | 0.150 ± 0.011 | 0.244 ± 0.012 | 0.356 ± 0.019 | 0.218 ± 0.009 |
| | | Node | | 0.151 ± 0.011 | 0.247 ± 0.011 | 0.364 ± 0.018 | 0.221 ± 0.009 |
| | DistMult | Overall | 0.906 ± 0.033 | 0.062 ± 0.018 | 0.129 ± 0.025 | 0.333 ± 0.013 | 0.127 ± 0.022 |
| | | Node | | 0.077 ± 0.023 | 0.169 ± 0.029 | 0.387 ± 0.008 | 0.159 ± 0.021 |
| | ComplEx | Overall | 0.996 ± 0.001 | 0.097 ± 0.037 | 0.193 ± 0.070 | 0.296 ± 0.066 | 0.161 ± 0.045 |
| | | Node | | 0.098 ± 0.037 | 0.195 ± 0.070 | 0.298 ± 0.065 | 0.163 ± 0.044 |
| x10.3 speed-up | RotatE | Overall | 0.996 ± 0.000 | 0.304 ± 0.015 | **0.491 ± 0.023** | **0.604 ± 0.016** | **0.412 ± 0.018** |
| | | Node | | 0.305 ± 0.015 | **0.493 ± 0.023** | **0.607 ± 0.016** | **0.414 ± 0.018** |

Table 6: All computed link prediction metrics (higher is better), with community prediction metrics also given separately.

| Dataset | Model | Value | Accuracy | F1 | ROC-AUC | AP |
|---|---|---|---|---|---|---|
| FB15k-237 | TransE | Community | 0.946 ± 0.000 | 0.941 ± 0.000 | 0.964 ± 0.000 | 0.812 ± 0.005 |
| | | Overall | 0.895 ± 0.001 | 0.896 ± 0.001 | 0.940 ± 0.000 | 0.727 ± 0.003 |
| | DistMult | Community | 0.864 ± 0.007 | 0.877 ± 0.006 | 0.970 ± 0.004 | 0.861 ± 0.016 |
| | | Overall | 0.908 ± 0.002 | 0.912 ± 0.001 | 0.950 ± 0.001 | 0.841 ± 0.005 |
| | ComplEx | Community | 0.997 ± 0.000 | 0.997 ± 0.000 | 0.998 ± 0.000 | 0.996 ± 0.000 |
| | | Overall | 0.938 ± 0.004 | 0.938 ± 0.003 | 0.968 ± 0.000 | 0.900 ± 0.002 |
| x4.4 speed-up | RotatE | Community | 0.988 ± 0.001 | 0.989 ± 0.001 | 0.998 ± 0.000 | 0.995 ± 0.000 |
| | | Overall | 0.925 ± 0.001 | 0.922 ± 0.001 | 0.960 ± 0.001 | 0.853 ± 0.003 |
| WN18RR | TransE | Community | 0.989 ± 0.000 | 0.989 ± 0.000 | 0.986 ± 0.000 | 0.963 ± 0.006 |
| | | Overall | 0.905 ± 0.002 | 0.904 ± 0.001 | 0.923 ± 0.001 | 0.798 ± 0.005 |
| | DistMult | Community | 0.882 ± 0.003 | 0.893 ± 0.002 | 0.999 ± 0.000 | 0.998 ± 0.000 |
| | | Overall | 0.920 ± 0.002 | 0.913 ± 0.002 | 0.860 ± 0.003 | 0.752 ± 0.005 |
| | ComplEx | Community | 1.000 ± 0.000 | 1.000 ± 0.000 | 1.000 ± 0.000 | 1.000 ± 0.000 |
| | | Overall | 0.925 ± 0.001 | 0.917 ± 0.001 | 0.912 ± 0.001 | 0.814 ± 0.001 |
| x4.6 speed-up | RotatE | Community | 0.997 ± 0.000 | 0.997 ± 0.000 | 0.999 ± 0.000 | 0.999 ± 0.000 |
| | | Overall | 0.912 ± 0.000 | 0.900 ± 0.000 | 0.949 ± 0.001 | 0.871 ± 0.001 |
| NELL-995 | TransE | Community | 0.995 ± 0.000 | 0.995 ± 0.000 | 0.992 ± 0.002 | 0.982 ± 0.002 |
| | | Overall | 0.932 ± 0.001b | 0.933 ± 0.001 | 0.963 ± 0.001 | 0.818 ± 0.009 |
| | DistMult | Community | 0.950 ± 0.003 | 0.952 ± 0.003 | 0.996 ± 0.001 | 0.978 ± 0.007 |
| | | Overall | 0.943 ± 0.002 | 0.943 ± 0.002 | 0.931 ± 0.002 | 0.836 ± 0.006 |
| | ComplEx | Community | 0.999 ± 0.000 | 0.999 ± 0.000 | 0.999 ± 0.000 | 0.999 ± 0.000 |
| | | Overall | 0.965 ± 0.001 | 0.965 ± 0.001 | 0.992 ± 0.000 | 0.963 ± 0.001 |
| x10.3 speed-up | RotatE | Community | 0.993 ± 0.001 | 0.993 ± 0.001 | 1.000 ± 0.000 | 0.999 ± 0.000 |
| | | Overall | 0.947 ± 0.004 | 0.945 ± 0.004 | 0.977 ± 0.001 | 0.885 ± 0.009 |

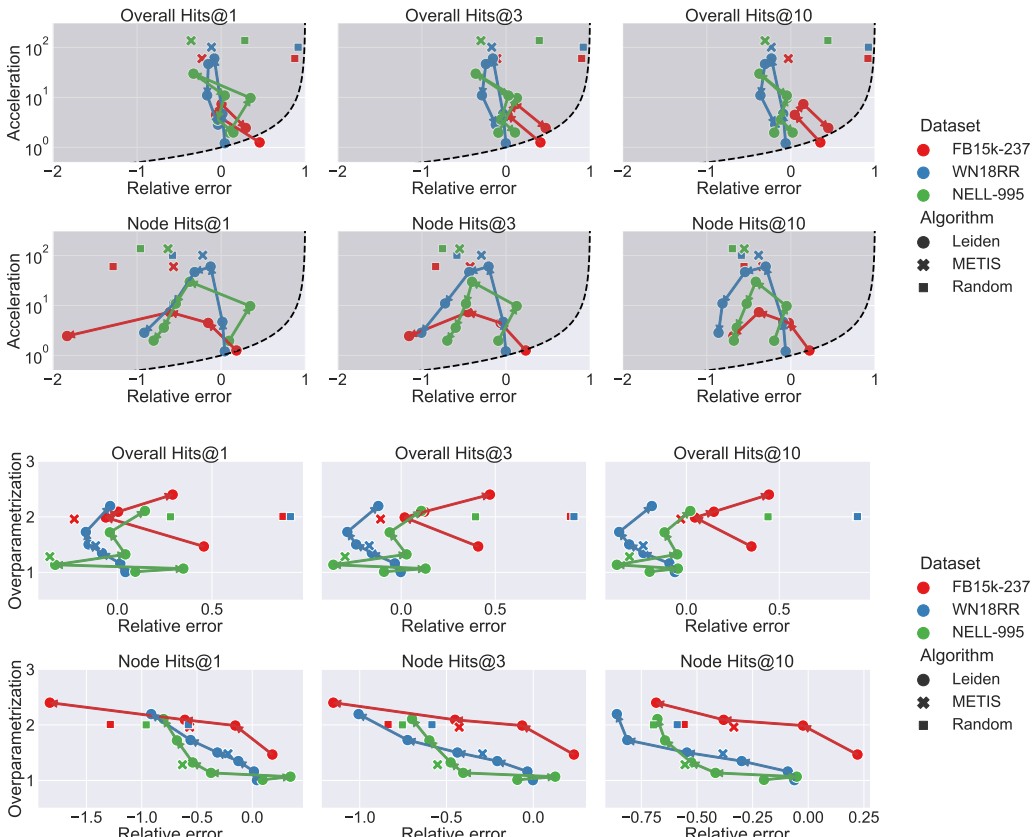

Figure 6: Plot of trajectories in the performance-scalability space, for the RotatE algorithm trained and evaluated with different community assignments. The Leiden paths are obtained by varying the resolution hyperparameter, the direction of the arrows indicates an increase of resolution. The feasible region for acceleration (Proposition 2) is the one shaded. Odd vs. even rows: COINs metric values after both steps vs. just the second step. Top two vs. bottom two rows: acceleration vs. overparametrization space.

## C.3   STABILITY & CONVERGENCE

The convergence plots in Figure 7 support our decision to model the final aggregate COINs loss value as the linear combination of the community and node terms. From the convergence lines, one can observe that the community and node iterates seem to converge at equal rates in both training and validation data. As such, the aggregate COINs loss function can be a simple average of the two terms without affecting convergence.

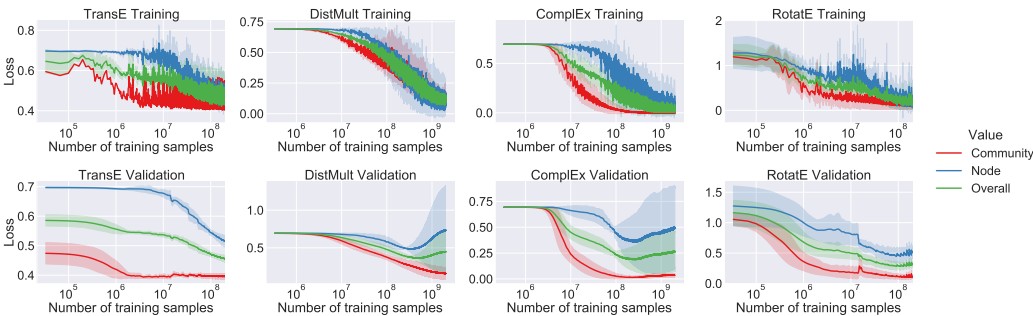

Figure 7: Plots of contrastive loss convergence over time, with the decomposition of the final COINs loss into the two constituent community and node loss terms. Top vs. bottom: training vs. validation loss. Left to right: the different COINs-integrated algorithms. Error bands show standard deviation across datasets.

## D   IMPLEMENTATION DETAILS

The entire implementation was performed in the Python 3.6 programming language. The Pandas library by McKinney (2010) was helpful with its efficient preprocessing operations on tabular data. The iGraph library by Csardi & Nepusz (2006) was utilized for the implementation of most of the graph analysis and preprocessing steps, including executing the Leiden algorithm. For METIS, the official software implementation (Karypis & Kumar, 1997) was invoked through a Python wrapper. The entire model architecture (along with integration code for the publicly available implementations of the external embedders), training, and evaluation, were implemented using the PyTorch deep learning framework by Paszke et al. (2019) and the extension framework for graph neural network learning PyTorch Geometric by Fey & Lenssen (2019).

All code was executed on a single machine with the following specifications:

- Intel® Xeon® Gold 5118 12-core CPU @ 2.30GHz;
- NVIDIA Tesla P100-PCIE-16GB GPU;
- 64 GB RAM.

To facilitate reproducibility, our full code implementation is available at:
`https://github.com/ResearchWeasel/coins-iclr-2024`.

