# OpenReview forum: "COINs: Model-based Accelerated Inference for Knowledge Graphs"
_ICLR.cc/2024/Conference — Submitted to ICLR 2024_

### Official Review · Reviewer_5igT · 2023-10-31

**Soundness:** 2 fair
**Presentation:** 2 fair
**Contribution:** 2 fair
**Rating:** 3
**Confidence:** 3

**Summary:**

This paper proposes a clustering-based approach to accelerate knowledge graph inference tasks such as link prediction. The basic idea (like prior work) is there must be a clustering of entities that makes most of the relations intra-cluster. The proposed method learns embeddings for clusters and nodes, and at inference time compares the embedding for the query node to all cluster representatives, picks the best cluster and then compares against all the entities in that cluster.

The authors show significant speedup on some KG inference tasks with decent quality losses.

**Strengths:**

- Formalizes the clustering-based approach and shows that it has decent performance.
- Attempts at explaining when the new algorithm is helpful.
- The new method speeds up training because not all weights are updated in each training step.

**Weaknesses:**

Experiments are limited:
- More datasets should be evaluated, especially bigger ones like WikiKG90M-LSC.
- Trade-off between error, speedup and number clusters should be investigated.
- Static min-cut algorithms (outside the end-to-end training) could be compared.
- What's the relationship between cut size, speedup and performance?

Missing literature
- Minimum cut literature studies the problem of reducing the number of inter-cluster relations.
- tf-GNN is another scalable GNN framework which uses sampling for large datasets.

**Questions:**

- For dense graphs, the cut will be really poor. FB15K is the densest graph you considered. How dense is it per relation type? Is the cut quality poor here? Does that explain some of the results?
- Did you consider producing different clusters for different relation types? Are the "optimal" clusters correlated?
- What's the breakdown of speedup for training and inference?

---

> ### Author Response · Authors · 2023-11-21
>
> Thank you for your comments, we agree on the relevance of the proposed new experiments and also believe they will improve our work. Due to the limited page count, we could only showcase the most important results in the main text and deferred many helpful additional results to the appendix for the submission. Nevertheless, we have managed to generate new figures and have uploaded a new draft. We provide more details about these and answers to your questions below, but to summarise, new result discussion is available in appendix sections C1 and C2, while Figures 4, 5, and 6 are new or updated.
>
> 1. **"More datasets should be evaluated, especially bigger ones like WikiKG90M-LSC."**
>
> - We agree on the relevance of experiments with larger datasets. We selected the three we use as the most classical ones for the field, and ones with the most available results from previous work to compare with as baselines. In fact, the WikiKG90M-LSC dataset does not have a publicly available test set with ground truth yet, as it was used for a recent competition. Nevertheless, we have started COINs-RotatE runs with larger knowledge graphs such as ogbl-biokg, but due to the greater number of edges, we require more time to achieve convergence. We would have no problem releasing these results when they are ready.
>
> 2. **"Trade-off between error, speedup and number clusters should be investigated. Static min-cut algorithms (outside the end-to-end training) could be compared. What's the relationship between cut size, speedup and performance?"**
>
> - Thank you for the interesting analysis suggestion, we have managed to perform these experiments and obtain results. The full discussion is available in appendix sections C1 and C2, however here we summarize. We implemented the METIS min-cut algorithm (used for partitioning by DistDGL (Zheng et al. 2020)) as well as random uniform community assignment (used for partitioning by Pytorch-BigGraph (Lerer et al. 2019)), as alternatives to Leiden. We analyzed the cut sizes produced by all 3 (large range of resolution values for Leiden) in Figure 4. We then end-to-end trained and evaluated COINs-RotatE with all 3 community methods (a couple of resolution values for Leiden in ascending order) and plotted the relative error-scalability trade-off in Figure 6. The results can be summarized as follows: although the METIS and random communities might yield higher speed-ups, there are a lot of Leiden community assignments resulting in similar performance, and equally significant speed-ups but with significantly fewer parameters.
>
> 3. **"Minimum cut literature studies the problem of reducing the number of inter-cluster relations. tf-GNN is another scalable GNN framework which uses sampling for large datasets."**
>
> - Thank you for your helpful suggestions. We have added a comment on tf-GNN in our introduction and corrected the error w.r.t min-cut literature (mention of the min-cut problem and reference to the METIS algorithm added in Section 2.3.3 Edge locality, which as mentioned above, we now consider METIS as an alternative method for obtaining the community partition and for which we obtained some initial experiment results).
>
> 4. **"For dense graphs, the cut will be really poor. FB15K is the densest graph you considered. How dense is it per relation type? Is the cut quality poor here? Does that explain some of the results?"**
>
> - If we assume that there can be only one relation between two entities, FB15k-237 has 0.2934% percent of all possible edges present. This number is significant only when compared to the densities of WN18RR and NELL-995, which are 0.0111% and 0.0054% respectively, so at least 25 times denser than both. Thus yes, unfortunately, it was the hardest to obtain good community splits for FB15k-237 (max modularity achieved lower than 0.5, while could easily obtain 0.75-0.8 for the other two) and yes, the bad community split was the reason for the very bad results of TransE and DistMult on Freebase and we give a bit more details on this via Table 5 in the appendix too.

---

> > ### Author Response · Authors · 2023-11-21
> >
> > 5. **"Did you consider producing different clusters for different relation types? Are the "optimal" clusters correlated?"**
> >
> > - No, as we believe this would cause too large of an increase in the number of parameters to be appropriate for our scalability goals in any case ($|R|$ new sets of 1 community embedder, $K$ intra-community embedders and 1 inter-community embedder), when the number of possible relations $|R|$ is large, but most relations could be underrepresented and we cannot assume any correlation between relation frequencies between community subgraphs. Instead, we consider handling the adaptivity to different community structures per relation by allowing each of the new COINs embedders to keep separate embeddings of each relation. This yields an addition of only $|R|(K + 2)$ new embeddings, and since $|R| \ll |V|$, the scalability effects for relation embeddings would be minuscule compared to those of node embeddings we analyzed thoroughly.
> >
> > 6. **"What's the breakdown of speedup for training and inference?"**
> >
> > - The speed-up for inference for each dataset is given in Table 2 in the Acceleration column, based on the ratio between the baseline complexity $N |V|$ and our complexity detailed in Proposition 1. For training, one can observe that without COINs in a worst-case training batch, one would have to compute all $|V|$ node embeddings and their gradients, while for COINs this worst-case number is instead (previously in our response to Reviewer 2jBS's related question there was an erroneous factor of $B$): $K+\max{(\max_{k=1}^{K}{|C_k|},|V^*|+1)}$, where similarly to the inference cost it depends on quantities that we all aim to optimize with better community partitions. The specific reduction in node embeddings per mini-batch for every dataset would be: 1.0075x for FB15k-237, 3.1450x for WN18RR, and 3.9284x for NELL-995. For FB15k-237 the number of inter-community nodes is what kills the training gains (due to this graph's high edge density), while for the other two, the size of the largest community dominates.

---

> > > ### Comment · Reviewer_5igT · 2023-12-01
> > >
> > > Thanks for the responses and additional experiments. I think you are on a good trajectory with this paper. At this point, I will raise my score from 3 to 4 but I still believe the paper should be polished, have more extensive experiments, and review the relevant work more comprehensively.

---

### Official Review · Reviewer_zPEm · 2023-11-05

**Soundness:** 3 good
**Presentation:** 3 good
**Contribution:** 2 fair
**Rating:** 6
**Confidence:** 2

**Summary:**

This paper studies a graph representation that takes clustering/coarsening information into account. The proposed algorithm partitions the vertices into a smaller number groups, and builds the embedding from a combination of inter and intra cluster objectives. The paper gives some bounds on the performances of this scheme, and experimentally shows a speed up factor of 4~10 plus slight improvements in prediction qualities.

**Strengths:**

The proposed scheme is natural, and the performance gains obtained are significant. Most other high performance embedding schemes I'm aware of directly go to continuous / geometric representations. Having a more graph theoretic intermediate stage feels useful for both designing faster algorithms and better leveraging graph structures.

**Weaknesses:**

The theoretical justifications are mostly limited to the running times, and don't seem to go into details about why the prediction estimates obtained are also better.

The data sets for the experiments are a bit different than those used in the graph embedding literature. For me it was a bit difficult to compare the experimental results here with other embeddings that I'm familiar with. More context on how to make such comparisons, especially on the importance of these benchmarks in the knowledge graph literature, would have helped a lot.

**Questions:**

As someone unfamiliar with knowledge graphs (my backgrounds are more in optimization / numerical methods), a direct comparison of the overall objectives optimized in the Leiden algorithm and this algorithm would be quite helpful. Otherwise I was only able to piece together the overall objective function from the pseudocodes, and had difficulties identifying the (black box) interactions with it.

---

> ### Author Response · Authors · 2023-11-18
>
> Thank you for your comments and support for our work. We have answered the few questions you had below.
>
>
> 1. **"The theoretical justifications are mostly limited to the running times, and don't seem to go into details about why the prediction estimates obtained are also better."**
>
> - We chose to focus our theoretical analysis on the scalability effects, as we deemed these to be our main contribution, but we would be for sure very interested for future work to provide generalization guarantees. This task was deemed out of our research scope and we predict it would be very difficult: one would have to at least merge the representation theory results for all of the knowledge graph algorithms we considered, with our algorithm formulation and integration of communities into the representation. One can have nevertheless some crude intuition: using the principle of locality for prediction simply makes the task a bit easier (given good communities), as both the number of possible answers and non-answers to a query greatly fall in comparison when predicting over the whole graph.
>
> 2. **"The data sets for the experiments are somewhat less well-known: so from the paper itself it's a bit hard to compare this approach with other embeddings (although there literature there is huge)."**
>
> - In fact, the datasets that we have chosen for evaluation are *specifically* the most standard for knowledge graph reasoning in an academic setting. The Freebase and Wordnet graphs appear in all of the original papers for TransE, DistMult, ComplEx and RotatE in some form (only Bordes et al. in 2013 used for TransE the versions with "inverse relation test leakage"), while NELL-995 is more common in GNN-based knowledge graph reasoning research (e.g. KBGAT). We reference all of these works.
>
> 3. **"As someone unfamiliar with knowledge graphs (my backgrounds are more in optimization / numerical methods), a direct comparison of the overall objectives optimized in the Leiden algorithm and this algorithm would be quite helpful: right now I'm only able to piece together the overall objective function from the pseudocodes."**
>
> - We apologize for the confusion. To clarify, we do not replace the Leiden algorithm for community detection, in fact, we utilize it without modification (as a blackbox implementation) to obtain communities as a graph preprocessing step. The Leiden objective is the Constant-Potts-Model version of modularity and its optimization simply aims to yield better community partitions. On the other hand, given a community partition, COINs modifies the contrastive learning process of a given embedding algorithm by integrating the embedding of those communities into its objective.

---

### Official Review · Reviewer_2jBS · 2023-11-07

**Soundness:** 1 poor
**Presentation:** 2 fair
**Contribution:** 2 fair
**Rating:** 5
**Confidence:** 4

**Summary:**

The draft presents a method to accelerate the knowledge graph single-hop tail query answering by hierarchical prediction from community detection. It shows that predicting relations between communities/clusters can be pretty accurate (for some embedding methods). So, to predict a tail query, we can avoid querying every node and instead predict the cluster and the node inside the cluster. Further, the authors decompose the embedding to model into "inside-cluster," "outside-cluster," and "inter-cluster" and blend them in the loss function. They show that the proposed method is pretty promising: when well-configured, it didn't decrease the evaluation metrics and is times faster than the naive all-node prediction.

**Strengths:**

The draft verifies the assumption that cluster hierarchical prediction may be much easier than node prediction and has commercial potential to accelerate tail query answering. It compares multiple embedding methods in the framework and does a set of ablation studies on the hyper-parameters, including the resolution parameter used in the modularity maximization. It is good to see that the community detection algorithm works in knowledge graph domains. Hierarchical prediction methods are well-known and sometimes required in extreme classification and nearest-neighbor queries. In knowledge graph prediction, it's mostly considered an engineering hack, but the draft verifies the assumption in the selected datasets.

**Weaknesses:**

However, the paper is not polished enough in mathematical rigor, typos, and organization. More possibly, it wasn't proofread before a hasty conference submission. This happens, but there are too many bugs to fix. And even if we remove all the mathematical-related parts, the experiments need to be stronger to be a pure evaluation paper. Thus, I suggest to reject the draft, and the detailed weakness is listed below.

First, The proposition doesn't prove the author's remark. For proposition 1 (equation 2), proving the lower bound on runtime didn't prove that your algorithm is better. I may show a lower bound of zero, and it says nothing. You need to prove the tighter upper bound for your optimized cluster size. The current upper bound is simply trivial, and I see it's possible to make your lower bound an upper bound (just substitute the values). For proposition 2, equation 3 simply moves the left-hand side to the right-hand side. You need to specify the scenario and quantify the "expected time to a correct answer" in your proposition. Also, I need clarification on why the derivation is related to Prop 1 (with a missing constant 2). It should be simply "Our method is better when ratio A is better than ratio B" in the derivation.

Second, you cannot control the cluster size of the community detection algorithm. Tuning the resolution parameter changes the number of clusters, but the size of each cluster depends on the graph structure and cannot be easily homogenized. There might be communities of a few nodes, and there might be a community consisting of 1/4 of the nodes. So, the analysis is actually "acceleration at the best case." The result is okay from a practical perspective, and showing good acceleration results is good enough.

Finally, the experiments don't support your claims in the introduction. All the 3 traditional datasets the authors tried can be run within hours or minutes on a single desktop. The experiments did show acceleration (in terms of vector evaluations), but the result doesn't support scalability compared to the scales in experiments from DistDGL or SMORE.

**Questions:**

Major questions:
1. (Algorithm 2 line 11) The $L$ function is actually implicitly parameterized by the graph structure and the negative-sampling methods. However, there are now three graphs: the community graph the intra- and inter-community graph. So when will each be used in the loss function? And what's the difference in sampling? For example, the node $\omega$ won't appear in the testing set but has many edges. How are they integrated?
2. (item 4 in page 5) Usually, the loss function is not convex to the embeddings. And there's no info on the refinement used.
3. (item 2 in page 5) The big-O notation is wrong. We always need to sweep through all embeddings.
4. (Sec 2.3.3 must have... be minimal) Modularity maximization (the Leiden method) doesn't purely minimize the inter-group edges.
5. (Sec 3.1) What's the purpose of adding the $\omega$ node?

Minor issues:
1. (Definition 3) citation.
2. (Definition 4) $\subseteq$ instead of $\in$
3. (Def 4 & 5) The "maximized" argument conflicts with the loss function.
4. (Sec 2.3.1) citations.

---

> ### Author Response · Authors · 2023-11-18
>
> We are grateful for your detailed comments and have implemented your suggestions for the text. The answers to your questions are below.
>
> 1. **"First, The proposition doesn't prove the author's remark. For proposition 1 (equation 2), proving the lower bound on runtime didn't prove that your algorithm is better. I may show a lower bound of zero, and it says nothing."**
>
> - We do not wish to claim that the proven lower bound is sufficient evidence for superiority. We do not base our further analyses on it. It was presented as simply the limit of what can be possible with COINs. It is very important that we could achieve it and thus the lower-bound is tight. As a result, we do not need to worry about a vacuous lower-bound of 0.
>
> 2. **"The current upper bound is simply trivial, and I see it's possible to make your lower bound an upper bound (just substitute the values)."**
>
> - Yes, we agree that the proof of the bounds is less rigorous than desirable. Thus, we amended this error the following way: we now show first with Proposition 3 that an extreme value of the evaluation cost is only achieved if the community sizes and/or evaluation edge counts per community are uniformly distributed. Then, we show with Proposition 4 that the value of $K$ is what decides whether the extreme point is a maximum or minimum.
>
> 3. **"For proposition 2, equation 3 simply moves the left-hand side to the right-hand side. You need to specify the scenario and quantify the "expected time to a correct answer" in your proposition. Also, I need clarification on why the derivation is related to Prop 1 (with a missing constant 2). It should be simply "Our method is better when ratio A is better than ratio B" in the derivation.""**
>
> - Thank you for the helpful suggestion, we have both simplified and clarified better the reasoning behind Proposition 2.
>
> 4. **"Second, you cannot control the cluster size of the community detection algorithm. Tuning the resolution parameter changes the number of clusters, but the size of each cluster depends on the graph structure and cannot be easily homogenized. There might be communities of a few nodes, and there might be a community consisting of 1/4 of the nodes. So, the analysis is actually "acceleration at the best case." The result is okay from a practical perspective, and showing good acceleration results is good enough."**
>
> - Yes we agree on the validity of your claim and counterexample. But to clarify, Proposition 1 holds for *any* disjoint partition $c$ and does not assume that Leiden community detection was used to obtain it. One can easily assign the groups arbitrarily focusing only on their size. E.g., with $c$=`np.random.choice(`$K$`, size=`$|V|$`)` just like what Lerer et al. in 2019 employ for PyTorch-BigGraph, one achieves pretty uniformly sized communities with low enough variance in the sizes. The idea is that due to this lack of correlation in such an assignment, the community prediction step for us would be extremely difficult.
>
> 5. **"Finally, the experiments don't support your claims in the introduction. All the 3 traditional datasets the authors tried can be run within hours or minutes on a single desktop. The experiments did show acceleration (in terms of vector evaluations), but the result doesn't support scalability compared to the scales in experiments from DistDGL or SMORE."**
>
> - We chose those datasets as most, if not all, previous work has produced baselines for us to compare with. More importantly, we do not claim to replace DistDGL or SMORE in all application settings. The key gaps in the research that we address is the *requirement of multiple machines* for lossless scalable prediction, as well as the lack of scalability improvements for knowledge graph model *inference* and not simply faster training data sampling and model optimization. If one has at their disposal compute for parallelization and has to guarantee no deviation from the baseline prediction power, the other frameworks are viable. As a side note, we utilize SMORE's improved training query and answer sampling too, while also noting the potential in parallelizing the intra-community embedding of nodes for future work.

---

> > ### Author Response · Authors · 2023-11-18
> >
> > 6. **"(Algorithm 2 [sic] line 11) The function is actually implicitly parameterized by the graph structure and the negative-sampling methods. However, there are now three graphs: the community graph the intra- and inter-community graph. So when will each be used in the loss function? And what's the difference in sampling? For example, the node won't appear in the testing set but has many edges. How are they integrated?"**
> >
> > - We apologize for the lack of clarity. Thus, we give a bit more implementation details about Algorithm 1: to learn better community representations negative community examples have to be sampled according to the community graph as well. Using these and the positive community triplets from the batch first the loss terms $\mathcal{L}(e^{(c)}_h,e^{(c)}_r,e^{(c)}_t,y)$ are computed in a supervised manner according to the contrastive loss over the community graph. Then the triplet batch is split into sub-batches according to the community membership of the entities and the appropriate intra- or inter-community embedder for each sub-batch computes the temporary triplet embeddings (superscripted with $^*$). Negative node samples are sampled from the graph appropriate for the target community of each sub-batch (that community's subgraph). Temporary triplet embeddings are then simply concatenated across the batch dimension before going to step 8.
> >
> > 7. **"Usually, the loss function is not convex to the embeddings. And there's no info on the refinement used."**
> >
> > - We apologize for the imprecise statement. The idea is that if a given model's embedding scoring function $\mathcal{S}$ is convex w.r.t. the embeddings (like for TransE), then the weighted averaging operations (steps 8-10 of Algorithm 1, which are the final refinement steps) will preserve the convexity of $\mathcal{S}$ w.r.t. the embeddings, and the contrastive loss is always convex.
> >
> > 8. **"The big-O notation is wrong. We always need to sweep through all embeddings."**
> >
> > - Yes, but that would be with respect to epochs, what we have is with respect to mini-batches. If we assume mini-batch stochastic optimization, COINs will have at most $K + B \max(\max_{k=1}^{K}{|C_k|},|V^*|+1)$ node embeddings or their gradients in memory for a given mini-batch size of $B$.
> >
> > 9. **"(Sec 2.3.3 must have... be minimal) Modularity maximization (the Leiden method) doesn't purely minimize the inter-group edges."**
> >
> > - Reviewer 5igT noticed this same error and we have implemented their suggestion of referencing min-cut literature.
> >
> > 10. **"What's the purpose of adding the $\omega$ node?"**
> >
> > - It is a padding node (embedding) representing all of the nodes in $V \setminus V^*$ in the eyes of the inter-community embedder. The idea is that if the knowledge graph model requires the sampling of random walk contexts, k-hop neighbors, multi-hop query structure etc. around an inter-community edge then the entity sampling could in these cases produce results outside of $V^*$, which $f_*$ would not know how to deal with. Through the inclusion of $\omega$ this is prevented for any possible scenario.

---

### Author Response · Authors · 2023-11-18

Many thanks to all reviewers for their comments, we have uploaded a new draft implementing everything except the results for Reviewer 5igT's new proposed experiments. When we have those results, we will answer Reviewer 5igT's questions and give details on any new tables/figures in our response to them.

Note that before the discussion period started we had finished some new experiments on our own side: running all the algorithms with COINs over all datasets now with 5 different random seeds. With these, we updated Tables 3, 5 and 6 to showcase mean and standard deviation for metrics, and Figure 3 to have error bars.

---

### Meta-Review · Area_Chair_4Aez · 2023-12-09

**Metareview:**

This paper deals with the problem of scaling one-hop inference in knowledge graph (KG) embedding models by first performing community detection (node clustering) in the KG and then using such communities as an abstraction to perform hierarchical classification. The proposed methodology, named COIN, has been applied to standard KGs for link prediction and compared against several KG embedding models used as baselines for COIN.

The reviewers appreciated the direction of COIN but at the same time highlighted some shortcomings in the current version of the manuscript. First, the way the first three sections are written oscillates between between overly pedantic to not rigorously explaining the methodology. Second, the proposed analysis is not rigorous enough and the bound derivations might need to be revisited. Third, a deeper analysis should be carried to highlight the impact of the chosen community detection algorithm and its trade-off with speedup and link prediction accuracy. I personally believe that results are potentially impressive, but as highlighted by reviewers, I agree that ablations and more rigorous statistical analysis are needed to fully support them. And a rewriting of the paper can turn it into a very strong contribution.

The paper is therefore rejected.

**Justification For Why Not Higher Score:**

The current presentation feels rushed, and would need some major rewriting (and possibly a couple additional experiments).

**Justification For Why Not Lower Score:**

N/A

---

### Decision · Program_Chairs · 2024-01-16

Reject